# STERapp: Semiautomatic Software for Stereological Analysis. Application in the Estimation of Fish Fecundity

Almoutaz Mbaidin [1], Sonia Rábade-Uberos [2], Rosario Dominguez-Petit [3], Andrés Villaverde [3], María Encarnación Gónzalez-Rufino [4], Arno Formella [4], Manuel Fernández-Delgado [1] and Eva Cernadas [1,*]

[1] Centro Singular de Investigación en Tecnoloxías Intelixentes da USC (CiTIUS), Universidade de Santiago de Compostela, Rúa Xenaro de la Fuente Domínguez, 15782 Santiago de Compostela, Spain; moutaz@mutah.edu.jo (A.M.); manuel.fernandez.delgado@usc.es (M.F.-D.)
[2] Instituto de Investigaciones Marinas, CSIC, Calle Eduardo Cabello 6, 36208 Vigo, Spain; soniaru@iim.csic.es
[3] Centro Oceanográfico de Vigo, Instituto Español de Oceanografía (IEO, CSIC), Subida a Radiofaro, 50-52, 36390 Vigo, Spain; rosario.dominguez@ieo.es (R.D.-P.); andres.villaverde@ieo.es (A.V.)
[4] Laboratorio de Informática Aplicada (LIA2), Departamento de Informática, Campus Universitario, Universidade de Vigo, Edifício Politécnico s/n, 32004 Ourense, Spain; nrufino@uvigo.es (M.E.G.-R.); formella@uvigo.es (A.F.)
[*] Correspondence: eva.cernadas@usc.es; Tel.: +34-604-032-096

**Abstract:** Stereology is the tridimensional interpretation of bidimensional sections of a structure, widely used in fields such as mineralogy, medicine, and biology. This paper proposes a general software to do stereological analysis, called STERapp, with a friendly graphical interface to enable expert supervision. It includes a module to estimate fish fecundity (number of mature oocytes in the ovary), which has been used by experts in fish biology in two Spanish marine research centers since 2020 to estimate the fecundity of five fish species with different reproductive strategies and oocytes characteristics. This module encloses advanced computer vision and machine learning techniques to automatically recognize and classify the cells in histological images of fish gonads. The automatic recognition algorithm achieved a sensitivity of 55.6%, a specificity of 64.8%, and an average precision of 43.1%. The accuracies achieved for oocyte classification were 84.5% for the maturity stages and 78.5% for the classification regarding presence/absence of the nucleus. This facilitates the analysis and saves experts' time. Hence, the SUS questionnaire reported a mean score of 81.9, which means that the system was perceived from good to excellent to develop stereological analysis for the estimation of fish fecundity.

**Keywords:** stereology; texture analysis; classification; support vector machine; software engineering; image segmentation; fecundity methods; oocytes; recognition; Weibel grid

## 1. Introduction

Stereology [1] is a corpus of techniques and methodologies to estimate unbiased quantitative geometrical parameters (for instance, number, length, area, and volume) from solid material, based on the relationship between a 3D structure and 2D cross sections obtained from it, i.e., stereology is the tridimensional interpretation of bidimensional sections of a structure [2]. Stereology uses random, systematic samples to estimate these parameters robustly and is widely applied in different scientific fields from mineralogy to neurology. It allows us to estimate, applying the Delesse principle, the number of particles and the volume they occupy within a structure. For that purpose, microphotographs of different tissues or materials are commonly used [3], assuming that they are thin enough to be considered a plane (2D).

Weibel and Gómez [4] developed a method for particles having a constant shape that is frequently used in biology and relates the number of particles per unit of volume (NV) to the number of particle shapes per unit of area in each section (NA). For that purpose, a

stereological analysis uses a grid of hexagonal cells (Weibel grid) whose cell size is adjusted to the size of the image and the objects (particles) to be measured. The grid is superimposed on the image to count the number of points (vertices of hexagonal cells) that are in contact with an individual particle and the number of particles included within the frame of the grid. Subsequently, each marked point is associated with the area of the corresponding hexagonal cell, where it is assumed that the number is proportional to the area of the particle in the tissue section. These areas are associated with the number of particles within the frame and used to estimate NA. Until now, this tedious work has been carried out manually, which is both time-consuming as well as inaccurate, because the area is based on the extrapolation to hexagonal cells, instead of the real area of the particle. In that sense, Emerson et al. [5] developed a method to estimate fish fecundity (i.e., the number of mature oocytes in the ovaries) based on the work of Weibel and Gómez [4], now largely used in fish biology research. Oocytes are spherical-shaped particles which increase in size and complexity as the gonadal maturation advances. Thus, applying stereological methods over histological sections of ovarian tissue requires experienced technicians capable of identifying the different stages of oocytes maturation in the histological slices.

Fecundity is a key parameter in the study of fish population biology and their dynamics. Its estimation [6] is used to calculate fish stocks' productivity as well as spawning stock biomass by applying egg production methods (EPM) in order to establish biological reference points that are used for sustainable fisheries management [7]. For example, currently EPM are used in the assessment of several commercial species targeted by the European fleets such as Atlantic mackerel (*Scomber scombrus*), Horse mackerel (*Trachurus trachurus*), European pilchard (*Sardina pilchardus*), or European anchovy (*Engraulis encrasicolus*).

Estimating fecundity by stereological methods implies a procedure to count and measure the diameters of oocytes (spherical cells) at different developmental stages (cortical alveoli, vitellogenic and hydrated), as well as those cells that will be reabsorbed (atretic) in histological images. For the diameter measurement, only those oocytes with a visible nucleus in the histological slide (i.e., cut closely through its maximum diameter) are considered. Despite stereology being a robust and unbiased method for estimating fecundity, it still requires specialized technicians and is time consuming, even with software being available today.

Current advances in computer vision and machine learning models may provide new tools to automatically obtain more accurate and more precise estimates of the stereological parameters, hence, improving the efficiency in routine laboratory work. ImageJ is the most popular tool to quantify particles from images in the biomedical field [8,9]. It provides many common computer vision techniques to process the images before quantification and it allows plugins to be defined that may implement specific methods or customize processing routines. However, in its current configuration, ImageJ does not allow object outlines to be corrected manually in a versatile and easy way before starting the quantification. The software Govocitos [10] proposed an integrated solution to calculate fish fecundity using the Weibel grid [4,5], overcoming the review limitations of ImageJ through a friendly graphical user interface (GUI). However, Govocitos is quite hard to install and to maintain for a non-expert user due to its complexity, mainly caused by the use of a database for collaborative work and data sharing. CystAnalyser [11] is a recent software we developed to count and measure cysts in histological images of liver and kidney, which also overcomes some of the limitations of ImageJ. Here, we propose STERapp, an application to do stereological analysis being an evolution of Govocitos using the technology of CystAnalyser to simplify its design and to increase both its versatility and its modularity. STERapp retains the automatic image processing module of Govocitos that recognizes and classifies oocytes (reproductive cells) in histological images of fish ovaries. Furthermore, it includes a new design for the GUI to supervise the automatic recognition before the stereological quantification. The recognized objects are stored in XML (extensive markup language) files in order to share and review the recognition results, and the stereological analysis itself is stored in CSV (comma separated values) files to simplify its use in other tools. STERapp

has been used since 2020 by experts in fish reproductive biology from two Spanish research centers, the Institute of Marine Research of CSIC and the Oceanographic Center of Vigo of the IEO. The main contributions of STERapp software are: (1) its friendly GUI simplifies the experts daily work flow to estimate the fish fecundity; (2) the automatic image processing is fast enough to operate in real time; (3) the experts can review and modify the results of the automatic oocytes recognition and classification before the image quantification step, overcoming other existing software packages such as ImageJ; (4) it allows export of joined results of various images together; (5) it allows training of the classifiers for different species; and (6) the software is accurate, trustworthy, and easy to install for non-expert users. The STERapp software is available from https://citius.usc.es/transferencia/software/sterapp (accessed on 7 June 2021) for Windows- and Linux-based systems for research purposes. The repository contains annotated images including their ground truth which was also used in this experimentation.

The rest of the paper is organized as follows. First, the architecture, the functionality and the automatic recognition and classification of objects of STERapp are described, along with the measurements used to evaluate the module of automatic processing to analyze the histological images of fish ovaries. Afterwards, the results are presented and discussed. Finally, the main conclusions are drawn.

## 2. Description of the STERapp Software

STERapp is a desktop application that can run on a general purpose computer under the operating systems Linux and Windows. It has been written in the C++ programming language. The GUI was developed using the GTK+ (GIMP Tool Kit) library (https://www.gtk.org (accessed on 7 June 2021)). Figure 1 shows the main window with an image loaded and displays the lateral panel of the module to analyze fish histological images. In the following sections we will describe the system architecture, expert perceptions of the functionality of STERapp, the automatic algorithms to recognize and classify cells, and the statistical evaluation of the automatic analysis.

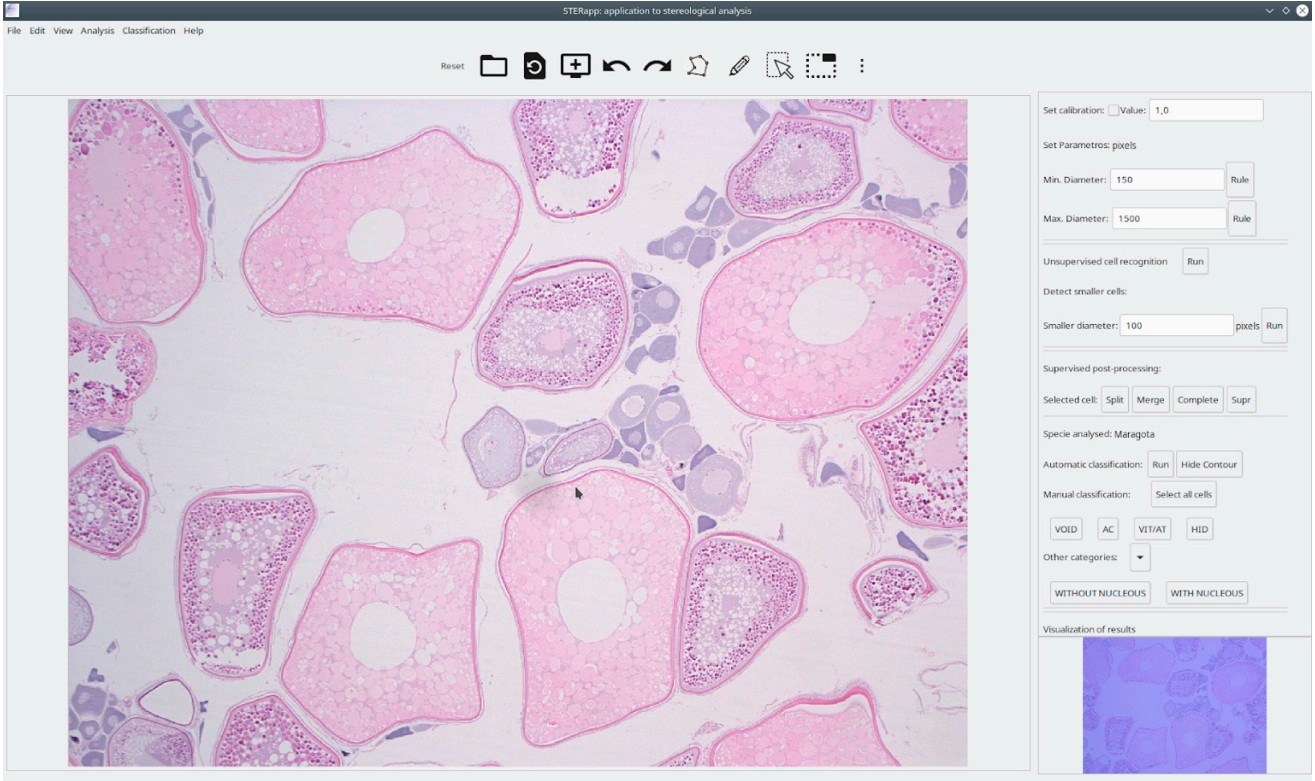

**Figure 1.** Main window of STERapp with a histological image of European hake ovary. On the right, we can see the full lateral panel showing the available functionality of the module to recognize and classify oocytes.

## 2.1. STERapp Architecture

STERapp is structured into two layers: the graphical user interface (GUI) and the application logic layer. The GUI layer provides a friendly interface to draw the outline of objects and manage them, and to interact with the software and visualize data. It is composed of four modules: (1) "preferences" configures the preferences of the software, including colors and widths of lines to draw the objects, calibration settings, minimum and maximum size of the objects to be recognized, categories of the objects to manage, and the fish species defined in the software; (2) "edit panel" manages the drawing, selecting, removing and, in general, manages the visualization of the objects and the results of the image analysis; (3) "analysis panel" allows the options to be selected in order to calculate the aggregated results of various images; and (4) "training panel" allows configuration of the classification training process. The user's guide supplied as supplementary material provides a detailed description of the graphical appearance and settings of these modules.

The application layer refers to the data processing, including the following modules, which will be described below: (1) "image segmentation" to automatically recognize objects in the image, i.e., cells in histological images of fish gonads; (2) "feature extraction" to extract numerical characteristics from a recognized object in the image; (3) "classification" to automatically train a classifier and classify objects; (4) "calculations" to compute the area and diameter of the objects; and (5) "data store" to save the calculated information into XML and CSV files. The overall application is modular and can be adapted to other stereological issues changing only the "image segmentation", "feature extraction", and "preferences" modules.

## 2.2. Functionality of STERapp

This section describes the main functionality provided by STERapp (flowchart of Figure 2). When we run STERapp the first time, we should configure the working preferences. One can work with default or preset configurations to recognize cells (oocytes) in the image, but we need to configure the species in order to train the classifier. By default, for the fish fecundity estimation, there are three maturation stages of oocytes configured in the preferences: cortical alveoli, vitellogenic, and hydrated, which correspond to the mature oocytes to be counted for in the fecundity calculations. STERapp allows setting preferences such as the location of the image and other files in the system, the calibration, the working colors for each stage, the definition of new categories, e.g., such as atresia (oocytes that are reabsorbed before spawning) and the species to analyze. After configuring the preferences, a common use case of STERapp starts loading an image in the view panel (Figure 1) and doing unsupervised cell recognition by clicking on the button "run" in the lateral panel. The detected cells are overlapped on the image in the drawing area. Due to the complexity of these types of images, the automatic cell recognition may not be perfect in 100% of the cases. Therefore, the user can review and supervise the outlines of the cells using the tools named "supervised post-processing" in the lateral panel, which allow the user to complete, split, and merge the objects having been automatically detected in the image. The user can also manually draw new cells or delete erroneously detected cells.

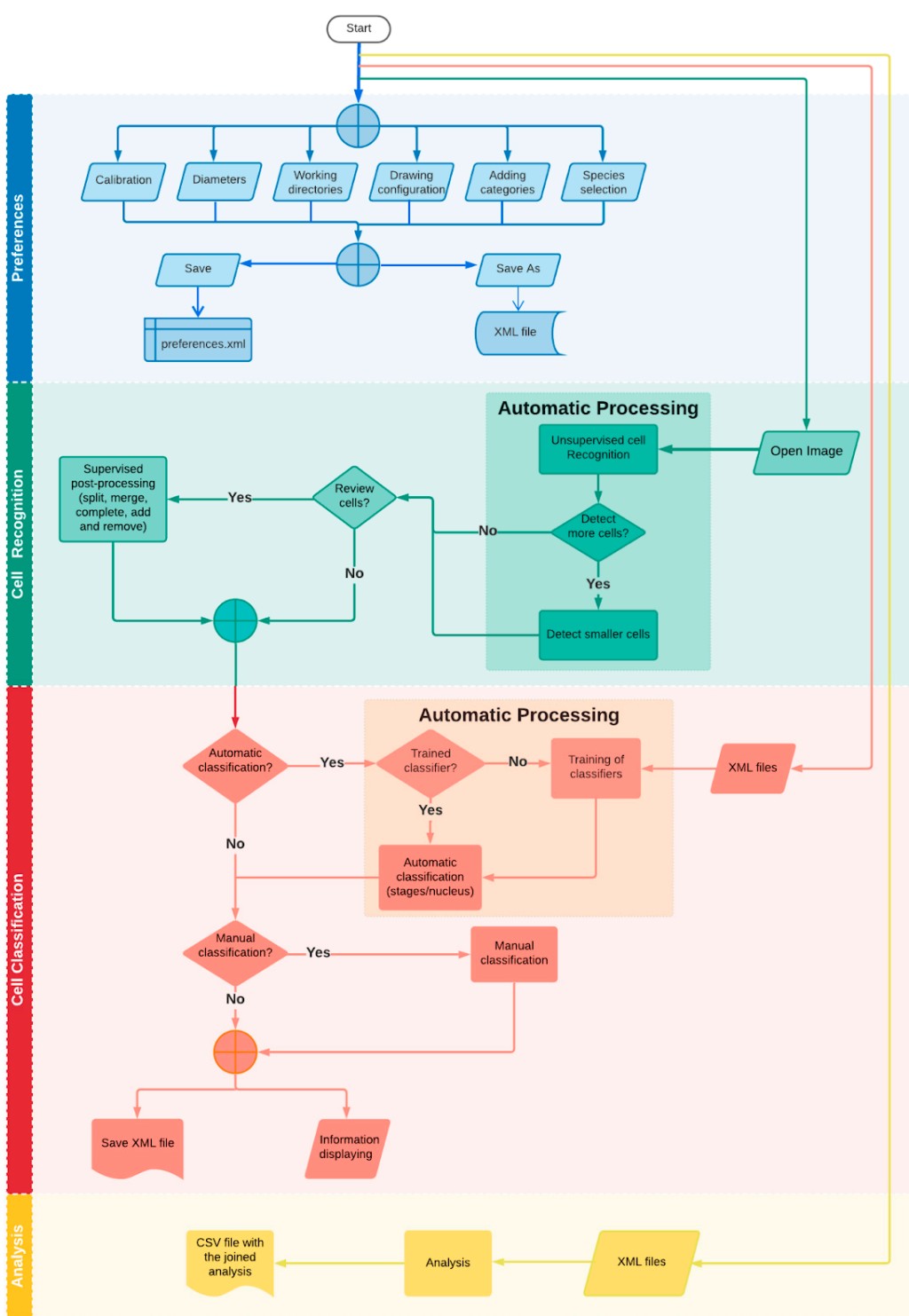

**Figure 2.** A flowchart containing the main tasks and functions of STERapp.

Once the outlines of the cells are satisfactory for the expert, the user runs the classifier in order to predict the categories of the objects. In our case, we use two classifications: (1) to discriminate oocytes with and without visible nucleus; and (2) to classify the three matured stages of the cells defined by default in the preferences. Again, if the automatic classification of cells is not suitable for the expert, the categories of the cells can be changed using the functionality after the label "manual classification" in the lateral panel. The expert can also set other categories to the cells defined previously in the preferences using the drop down list named "other category" in the lateral panel. Finally, once all the objects are recognized

and classified correctly after supervision by the expert, we can visualize the results or export them to XML and CSV files. The "visualization of results" tool automatically shows information of each cell drawn on the drawing panel, allowing the user to explore the area, diameter, or category, and view whether the cell must be counted or not to the stereological analysis in real time by clicking the "show table" tool. The cells can be highlighted in the image by selecting them in the table and, vice versa, selecting a cell in the image highlights its corresponding row in the table. In addition, STERapp also allows the user to load recognition results from XML files. The XML file contains the overlays drawn on the image, which can be later loaded at any time onto the image in order to check the image analysis, share it with a colleague, etc. Considering that in such fields as medicine or biology, there is quite a subjectivity in the analysis; a fact being very important in order to unify criteria among experts applying stereological analysis. The CSV file contains statistical information provided by quantitative analysis of the images: (1) spatial resolution, i.e., the relation between micrometers and pixels, at which the images were digitized (calibration), which must be pre-specified by the user; (2) image size in pixels; (3) cell index; (4) number of cells; and (5) the area, diameter, maturation stage, and if the cell has a visible nucleus or not. The user's guide provides a more detailed description.

### 2.3. Automatic Recognition and Classification of Cells

The recognition of cells in the image is a segmentation problem, which divides the image into objects (or cells) and background. STERapp uses a multi-scale algorithm based on the Canny filter [12] already used by Govocitos [10], with very good results with the European hake and Pouting species. Once the cells are recognized, the software classifies the cells according to: (1) the presence or absence of a visible nucleus in the cell (a two class problem); and (2) the development stage of the cell, considering only the three stages predefined in the preferences: cortical alveoli, vitellogenic, and hydrated. The classification is performed using the support vector machine (SVM), one of the strongest classifiers in the literature [13]. Two SVM classifiers, one for each type of classification, are used by STERapp, both of them with radial basis function (RBF) kernel implemented by the LibSVM library [14] and accessed through its C++ binding. The input of the classifier is a vector of features extracted from the image characteristics of the cell. Many features based on the image texture and color properties were tested in previous work [15]. For STERapp, we selected a combination of features that provided a good trade-off between the time required for the calculation and the performance achieved by the SVM classifier. Eventually, the feature vector is composed by: (1) the uniform rotation invariant local binary patterns (LBP) calculated on a grey-level version of the original image using irregular regions (10 features); and (2) the first order statistics of the color RGB image into the cell (15 features).

The SVM classifier must be trained before predicting the class for a cell (presence/absence of a visible nucleus or the development stage). STERapp allows the training of the SVM classifier for the fish species defined in the preferences. The module "training panel", through the submenu classification → train classifier allows the user to set the species and the XML files (generated by STERapp) that will be used to train the classifiers. These files must contain the objects recognized for a collection of images, along with their classes. The collection of images should be representative enough for the classification problem, and must contain objects of all classes. The next subsection describes the training.

### 2.4. Training of Classifiers

For the training, we consider a maximum number of 1000 cells randomly selected from the XML files provided by the user, with the same number of patterns for each class whenever possible (a minimum number of objects is required for a class to be included in the training). STERapp implements the K-fold cross-validation methodology, with K = 4, using training and validation sets to tune the two hyper-parameters of the SVM, i.e., the regularization ($\lambda$) and the RBF kernel spread ($\sigma$). The performance is evaluated by the

Cohen kappa statistic [16], which measures the agreement between the true and predicted category labels excluding the agreement by chance. Kappa (in %) is defined as:

$$K = 100 \frac{P_a - P_e}{s - P_e} \tag{1}$$

$$P_a = \sum_{i=1}^{C} C_{ii} \tag{2}$$

$$P_e = \frac{1}{N^2} \sum_{i=1}^{C} \left( \sum_{j=1}^{C} C_{ij} \right) \left( \sum_{j=1}^{C} C_{ji} \right) \tag{3}$$

$$S = \sum_{i=1}^{C} \sum_{j=1}^{C} C_{ij} \tag{4}$$

where $C_{ij}$ is the number of validation patterns that belong to class i and that are assigned by the SVM to class j; N is the total number of validation patterns. In each cross-validation trial, $K - 1$ folds (3 folds for K = 4) are used to train the classifier using each combination of hyper-parameter values, and the remaining fold is left for validation by evaluating kappa using the current combination of hyper-parameter values. This process is repeated K times, and the kappa on the validation fold is averaged over the K folds. The values of $\lambda$ and $\sigma$ used are: $\lambda = 2^i$ for i = −3 ... 12, and $\sigma = 2^i$ for i = −15 ... 0. Finally, each SVM is trained with the combination of hyper-parameter values showing the highest average kappa over the K validation sets using the entire data set, in order to classify new cells.

*2.5. Statistical Analysis*

To perform the statistical evaluation of the automatic recognition algorithms, STERapp registers the use of the software by the experts, saving additional information in the XML files. Let us define a true positive (TP) hit when a cell is correctly recognized and a false positive (FP) hit whenever the user manually deletes the cell using the GUI. A cell is considered false negative (FN) if the user manually adds it. Once we count the values for TP, FP, and FN for an image, we calculate the sensitivity (Se), specificity (Sp) and average precision (AP), in %, as:

$$Se = \frac{100 \, TP}{FN + TP} \tag{5}$$

$$Sp = 100 \left( 1 - \frac{FP}{FP + TP} \right) \tag{6}$$

$$AP = \frac{100 \, TP}{TP + FP + FN} \tag{7}$$

The automatic cell recognition algorithm of STERapp often provides detections close to those expected by the experts, who only use the "tools for supervision" (complete, split and merge) in order to fit the true cell contour. For example, the "complete" tool is used when the contour provided is close to the true cell contour, the "split" tool is used when two cells are detected together, and the "merge" tool is used when a cell is detected as several small pieces. In all these cases, the cells were detected, but their outline was not totally perfect. STERapp only registers the cells which are modified by these tools, and not the type of tool used in each case. Therefore, we define the ratio MC as the number of cells modified by the tools for supervision divided by the number of cells automatically recognized by the algorithm. As well, the scores Se, Sp, and AP are updated considering that the cells have been modified using "tools for supervision" are counted as true positive, because they are correctly detected and their outline is quasi-suitable (note that if the recognition were not almost perfect, the expert would delete the cell instead of modifying it).

## 3. Materials

In the present study, the analyzed images correspond to histological sections of ovaries of several species with different reproductive strategies and oocytes characteristics: Atlantic mackerel *(Scomber scombrus)*, Four-spot megrim *(Lepidorhombus boscii)*, Roughhead grenadier *(Macrourus berglax)*, Ballan wrasse *(Labrus bergylta)*, and European pilchard *(Sardina pilchardus)*.

Fish ovaries were extracted from fish and submerged in a solution of 4% buffered formaldehyde for a minimum of 7 days (or more depending on their size), to fix the cellular structures; subsequently, the ovaries were cut in slices and dehydrated by submerging them in an increasing concentration sequence of alcohols and organic solvents that allowed the infiltration of the tissue by paraffin or resin (only Atlantic mackerel), this process took from 12 to 24 h depending on the species. Finally, the slices of tissue were embedded in paraffin or resin, forming blocks that were sectioned in thin slices of 3–5 microns thick. These thin sections were laid in slides and stained with haematoxylin and eosin, as the gold standard stain. Pictures of ovary sections were taken with different image analysis systems at different magnifications (Table 1). The processing of tissues and analysis of images from different species, which were taken with different image systems and magnifications, increased the variability of oocyte shapes, sizes, aspects, abundance, and distribution within the image, which were used to test the robustness of the software.

**Table 1.** Characteristics of image analysis systems used to take pictures of ovary sections in each species.

| Species | Microscope | Camera | Magnification | Image Size |
|---|---|---|---|---|
| Atlantic mackerel | Nikkon Eclipse 80i | Nikkon DXM 1200F | 40× | 3840 × 3072 |
| Four-spot megrim | Leica DMRE | Leica DFC 320 | 2.5–10× | 2088 × 1550 |
| Roughhead grenadier | Leica DM 4000B<br>Leica M165C | Leica DFC 420<br>Leica DMC 4500 | 1.25×<br>0.73× | 3888 × 2916<br>2560 × 1920 |
| Ballan wrasse | Leica DMRE | Leica DFC 320 | 2.5–10× | 3132 × 2325 |
| European pilchard | Leica DMRE | Leica DFC 320 | 2.5–10× | 2088 × 1550 |

Figures 3–7 show examples of the different fish species studied, where the contours of the recognized cells and their classification were overlapped onto the image. The color shows the development stage of the cells: green for vitellogenic/atretic, yellow for cortical alveoli, and blue for hydrated. The type of line shows the presence/absence of a visible nucleus: continuous line for cells with a visible nucleus and dashed line for cells without a visible nucleus.

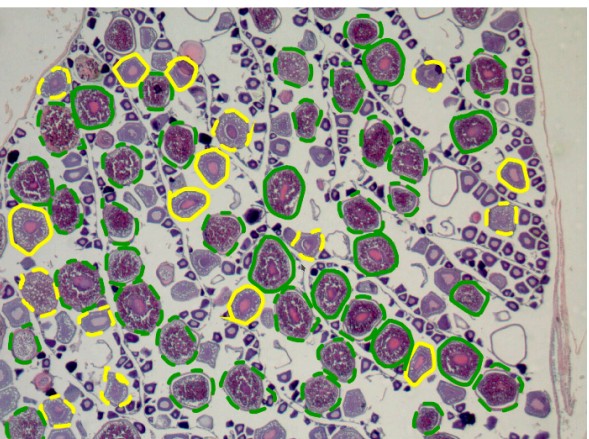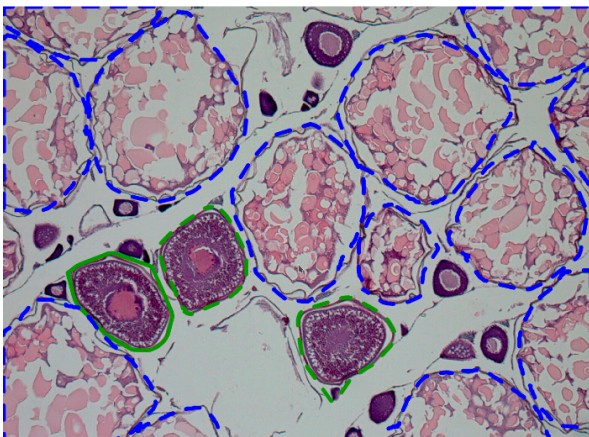

**Figure 3.** Examples of European pilchard ovary histological images with the contours of the cells supervised by the experts overlapped. The colors are used to distinguish oocyte developmental stages: green (vitellogenic/atretic), yellow (cortical alveoli), and blue (hydrated). A continuous line corresponds to cells with a visible nucleus and a dashed line to cells without a visible nucleus.

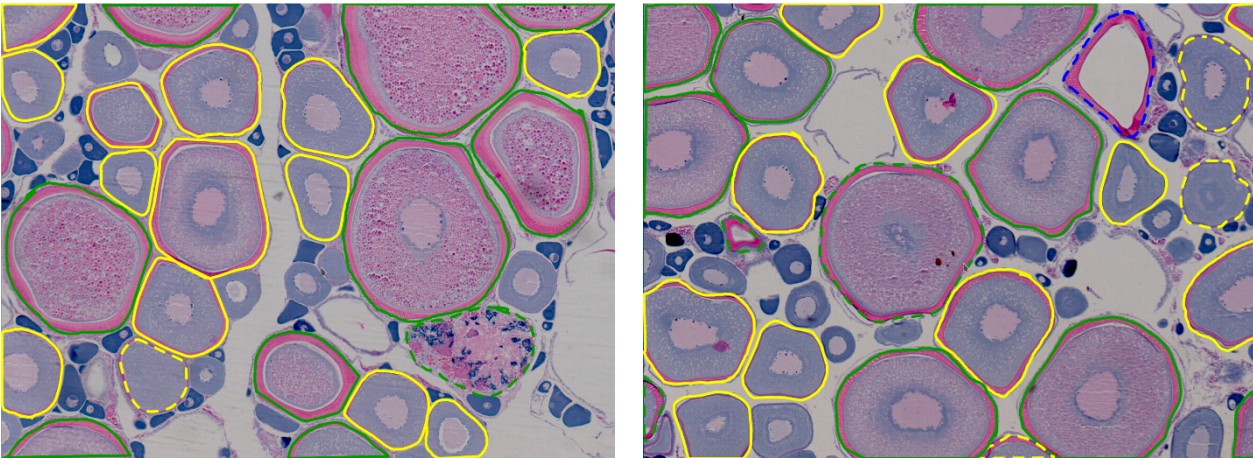

**Figure 4.** Examples of Ballan wrasse ovary histological images with the contours of the cells supervised by the experts overlapped. Line color and type have the same meaning as in Figure 3.

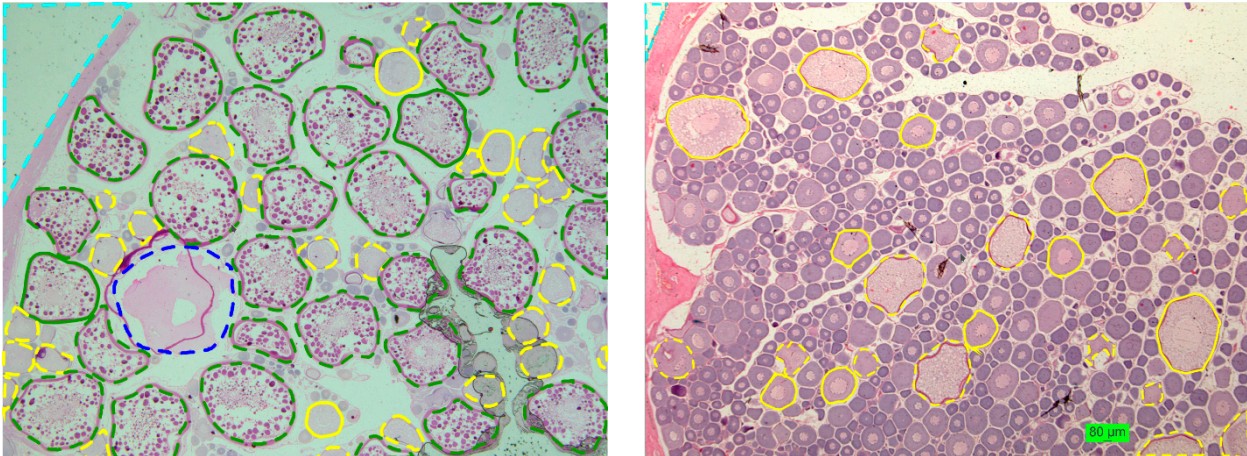

**Figure 5.** Examples of Roughhead grenadier ovary histological images with the contours of the cells supervised by the experts overlapped. Line color and type have the same meaning as in Figure 3.

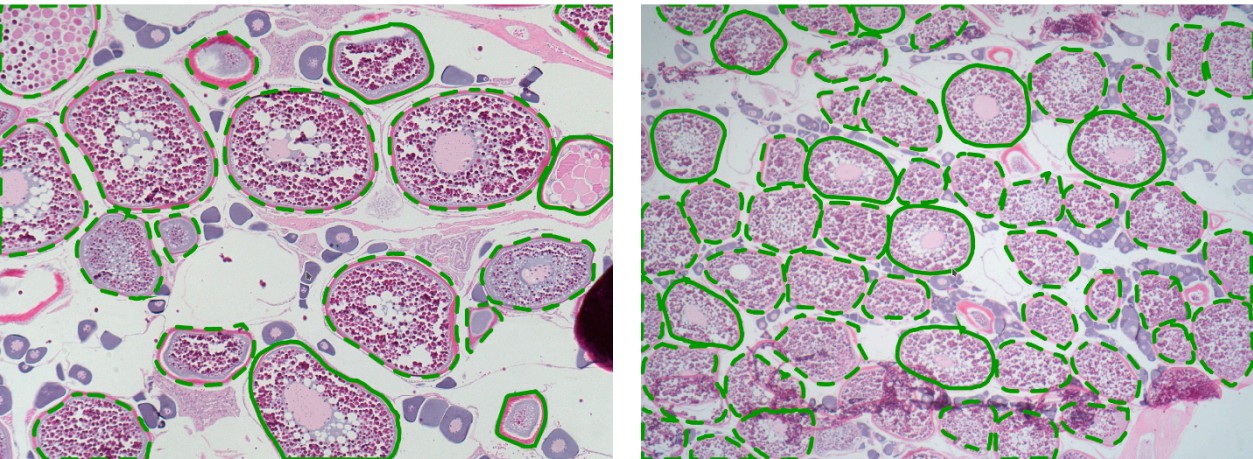

**Figure 6.** Examples of Four-spot megrim ovary histological images with the contours of the cells supervised by the experts overlapped. Line color and type have the same meaning as in Figure 3.

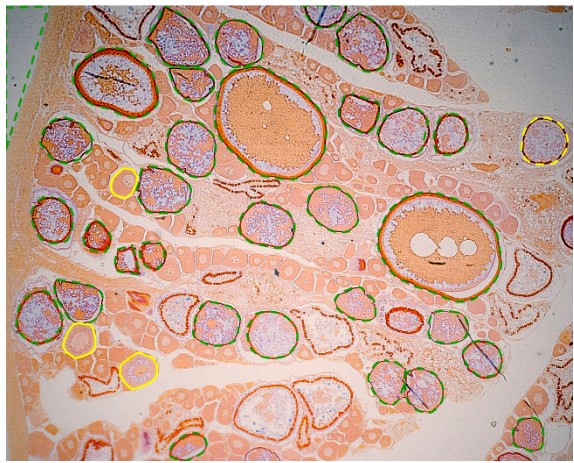 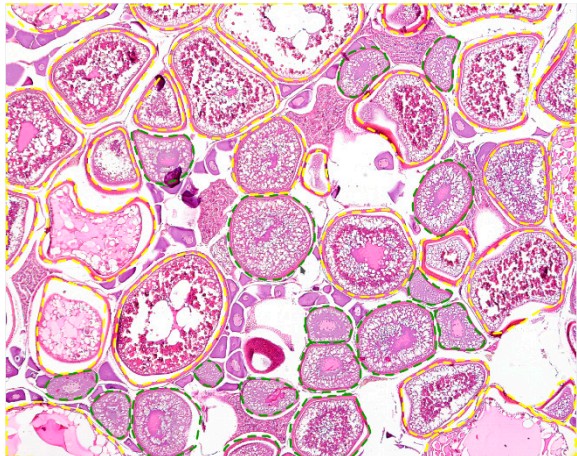

**Figure 7.** Examples of Atlantic mackerel ovary histological images with the contours of the cells supervised by the experts overlapped. Line color and type have the same meaning as in Figure 3.

## 4. Results and Discussion

Since 2020, STERapp was tested by expert technicians from the Instituto de Investigacións Marinas (IIM) and the Instituto Español de Oceanografía (IEO) from the State Agency Consejo Superior de Investigaciones Científicas (CSIC) with the aim to evaluate the software operating in a real environment. The biologists used STERapp in their daily work to do stereological analysis, for which they were required to recognize and classify the cells in the images. Their operations using STERapp were logged into XML files for later statistical evaluation of the automatic computer vision and machine learning algorithms incorporated in the software. The following subsections summarize the results achieved from different points of view: automatic detection and classification of cells, performance analysis, and perception of the system.

### 4.1. Automatic Detection and Classification

As our purpose is to evaluate the robustness and generalization capability of the software, STERapp was used in the fisheries labs with images of the five fish species listed in Section 3, which were acquired under different conditions of sample preparation and digitization (see Table 1). We analyzed 132 images: 27 of European pilchard, 20 of Four-spot megrim, 24 of Roughhead grenadier, 38 of Ballan wrasse, and 23 of Atlantic mackerel. The average number of cells by image is 35.8, varying from 22.9 in Ballan wrasse up to 44.8 in Four-spot megrim. Table 2 shows the results achieved for every species including sensitivity (Se), specificity (Sp), average precision (AP), and the ratio of modified cells (MC, in %).

The MC varies among species, ranging from 21.4% for Roughhead grenadier to 45.3% for European pilchard. As expected, Se, Sp, and AP increase significantly when the modified cells are considered as hits, normally more than 10 points. In this case, the average sensitivity, specificity, and average precision among species are, respectively, $55.6 \pm 8.2\%$, $64.8 \pm 16.2\%$, and $43.1 \pm 11.3\%$. These somewhat high standard deviations mean that there is a large difference among species. For example, the average precision ranged from 27.5% for Ballan wrasse up to 60.3% for the Four-spot megrim. This variability was also observed among the images of a given species, as we reported in our previous software, Govocitos, with the species European hake and Pouting [10]. This was one of the facts that led us to propose a semi-automatic software which allows the expert to supervise the cell recognition before starting the quantification and eventually estimate the fish fecundity.

**Table 2.** Performance of the automatic algorithm to recognize the cells in the image for all fish species studied. N and Nc are respectively the number of images and the average number of cells by image. Se, Sp, and AP are the sensitivity, specificity, and average precision, respectively. MC is the ratio (in %) of modified cells by using the supervision tools in the lateral panel. Se, Sp, and AP are calculated considering only the cells perfectly recognized and not modified by the expert (first row), and considering that the cells modified by the tools for supervision are correctly detected but their outlines were modified by the expert (second row). Average values in the last row are presented with the standard deviation (average ± SD) to give an insight into the variation among species.

| Species | N | Nc | MC | Se | Sp | AP |
|---|---|---|---|---|---|---|
| European pilchard | 25 | 36.5 | 45.3 | 39.8 <br> 57.8 | 70.5 <br> 80.3 | 34.8 <br> 50.5 |
| Four-spot megrim | 20 | 44.8 | 28.4 | 58.5 <br> 65.0 | 83.4 <br> 87.5 | 53.6 <br> 60.3 |
| Roughhead grenadier | 24 | 33.0 | 21.2 | 47.3 <br> 58.6 | 42.1 <br> 52.6 | 28.0 <br> 37.6 |
| Ballan wrasse | 38 | 22.9 | 23.1 | 26.3 <br> 40.3 | 31.5 <br> 45.6 | 18.8 <br> 27.4 |
| Atlantic mackerel | 23 | 42.0 | 35.0 | 46.1 <br> 56.1 | 43.8 <br> 58.2 | 33.4 <br> 39.5 |
| Average | 26.4 ± 6.2 | 35.8 ± 7.7 | 30.6 ± 8.7 | 43.6 ± 10.5 <br> 55.6 ± 8.2 | 54.3 ± 19.4 <br> 64.8 ± 16.2 | 33.7 ± 11.4 <br> 43.1 ± 11.3 |

Table 3 shows the average accuracy to classify the cells into the three development stages: cortical alveoli, vitellogenic, and hydrated, and into the presence/absence of a visible nucleus for each species. The average accuracy among species for the development stage is 84.8%, ranging from 58.5% for Ballan wrasse up to 98.7% for the Four-spot megrim. The average accuracy for the presence/absence of a visible nucleus ranged from 51.6% for Ballan wrasse, which comes close to random classification, up to 89.3% for the Roughhead grenadier, which is a very good score. These results agree with the study in [10], which used European hake and Pouting and where the development stage classification achieved better results than the nucleus classification. Ballan wrasse eggs have a characteristic thick chorion, a cellular coat surrounding eggs which protects them physically from the environment. This coat confounds the automatic recognition algorithm, which tends to recognize the outline of the cell as the interior margin of the chorion instead of the exterior. Furthermore, the extremely high affinity by eosin of the vitelline granules blur the lines between the nuclei and the vitelline granules. These two characteristics could explain the rather bad results obtained for this species.

**Table 3.** Average accuracy of the automatic classification for all fish species studied. The column "Stage" shows the accuracy for classifying the development stages: cortical alveoli, vitellogenic, and hydrated. The column "With/without nucleus" shows the accuracy to classify if the cell has a visible nucleus or not. Average values are presented with the standard deviation (average ± SD) to give an insight into the variation among species.

| Species | Stage | With/without Nucleus |
|---|---|---|
| European pilchard | 93.0 | 83.6 |
| Four-spot megrim | 98.7 | 86.5 |
| Roughhead grenadier | 95.1 | 89.3 |
| Ballan wrasse | 58.5 | 51.6 |
| Atlantic mackerel | 78.5 | 81.6 |
| Average | 84.8 ± 14.8 | 78.5 ± 13.7 |

## 4.2. Analysis Performance

In this subsection we compare the stereological analysis of an image using STERapp and the traditional method, based on the Weibel grid system [2]. This method uses a grid of hexagonal cells of which only the vertices are displayed, and further involves: (1) superimposing on the image the previous grid of dots separated by a distance equivalent to the size of the smallest particle to be measured, 100 microns approximately, for fish fecundity; (2) marking each of the dots on each object of interest (oocytes) to estimate the associated area; and (3) counting each object within the grid [5]. This work, besides being extremely tedious and visually uncomfortable, decreases the accuracy of the analysis, since the area occupied by each object is estimated based on the theoretical area of the hexagonal cell associated to the points over the object, i.e., it is not based on the real area of the object in the image, an advantage the present software exhibits. Moreover, STERapp uses the entire image (all pixels) for its calculations, whereas the Weibel grid, as it is commonly employed, does not usually occupy the entire image. The traditional Weibel grid system does not automatically classify the objects (oocyte maturity stages and with or without a nucleus), so it is the user who must associate the marked points and objects to each class manually, increasing the probability of making mistakes. In addition, the image correction tools provided by STERapp (merge, split, select, etc.) make the histological image analysis and detection of errors much easier, saving the contours and class labels of oocytes in XML files that allow a review or interruption of the work at any time. Similarly, the XML file format can be replicated, compared, or shared. The stereological analysis of the images can also be exported to CSV files, a file format easily suited to be managed in other tools. A further added value of STERapp is the opportunity to customize preferences and save them in XML file format for different users; this way, users from different locations can easily share preferences or work on the same computer in different work sessions. The simple installation and use of STERapp (see Section 4.3), its versatility for the creation of new object classes, as well as the ability to train its classifiers (in our case, for different species) also make it an application with great potential in other disciplines such as medicine or geology.

The time needed for processing histological images with STERapp depends on the size of the image and the computing power, as well as the characteristics of the ovarian structures of each species. We did not develop a formal statistical comparison of the time required by STERapp and the traditional Weibel grid system. However, the time required by experts is much shorter using STERapp because its intuitive and user-friendly GUI greatly facilitates technical work. Specifically, the time required by the automatic processing of one image is approximately one minute on a general purpose computer, plus the time required by the expert's supervision, which may range from 2 to 5 min, depending on the review needs. Comparatively, the traditional methods require about 10 min per image. Therefore, STERapp saves between $100(10 - 6)/10 = 40\%$ and $100(10 - 3)/10 = 70\%$ of time compared with the traditional method.

## 4.3. Expert Perceptions

We evaluated expert perceptions about STERapp using the system usability scale (SUS), a free questionnaire to measure the learning ability and subjectively perceived usability of computer systems [17,18]. This is a 10-item questionnaire with a five-point scale ranging from 1 (strongly disagree) to 5 (strongly agree), providing a final system score between 0 and 100. The score is calculated as follows: adding up the positively worded items (1, 3, 5, 7, and 9) and subtracting 1 from the user responses; adding up the negatively worded items (2, 4, 6, 8 and 10) and subtracting the user responses from 5. The SUS score for each user ranges from 0 to 40, which is multiplied by 2.5 in order to re-scale the value to the range from 0 to 100. This score is translated to people's rating of systems and products in terms of adjectives in order to provide a meaning to them [19] as follows: SUS < 25 is the worst imaginable system; from 25 to 39 is from the worst imaginable to poor; from 40 to 52 is from poor to OK; from 53 to 73 is OK to good; from 74 to 85 is good to excellent; and above 85 is excellent to the best imaginable system. Usually, a small sample (8–12 users) is

enough to give a good assessment of how people see the software. STERapp was evaluated using the SUS questionnaire by 8 expert technicians from the Instituto de Investigacións Mariñas (IIM) and the Instituto Español de Oceanografía (IEO). The mean score achieved was 81.9 (ranging from 67.5 to 95), which means "from good to excellent". These positive ratings are mainly associated with the ease of use, performance, and versatility of the software as highlighted in Section 4.2.

*4.4. Comparison with ImageJ*

As mentioned, ImageJ is the most widely used software for image analysis in the fields of biology and biomedicine [8,9]. ImageJ allows the development and installation of plugins to analyze a specific type of images. Nevertheless, the main drawback of ImageJ is that its GUI does not provide an easy way to correct or modify the automatic recognition of objects in the images before their quantification. Considering the complexity and variability of the histological images of fish gonads, we do not know of any approach that automatically recognizes oocytes in a completely satisfactory way for the majority of images, so enabling revision by experts is an essential part of any software. STERapp overcomes many limitations of ImageJ such as: (1) its friendly GUI provides an easy way for experts to supervise the fecundity calculation process; and (2) it allows the monitoring of each result through the information registered in the XML files and enables sharing of the analysis criteria among other experts. In fact, ImageJ was never used for fish fecundity estimation purposes due to its low usability, while STERapp was qualified by the users as "from good to excellent" with a mean score of 81.9 in the system usability scale.

## 5. Conclusions

This work proposes the software tool STERapp to do stereological analysis on images. It was evaluated for the analysis of histological images of fish gonads in order to estimate fecundity, a parameter commonly used in stock assessment for the sustainable management of fisheries. STERapp combines the automatic recognition and classification of cells in histological images with a friendly graphical user interface that allows the expert to modify the automatic cell recognition and/or classification before starting the quantification or the stereological analysis of the image. STERapp calculates the cell area and diameter and the number of cells in each development stage, which are the key parameters to estimate the fish fecundity. Furthermore, it checks if the cells are located on the image border in order to determine if they have been counted or measured in the stereological analysis. Since 2020, STERapp has been tested by two Spanish fisheries research groups that analyzed images of five fish species coming from different image acquisition systems. The automatic processing of the images is quite robust providing the following average performance over the five species studied: (1) cell recognition with a sensitivity of 55.6%, specificity of 64.8%, and precision of 43.1%; and (2) cell classification of 84.8% to discriminate three development stages and 78.5% to discriminate whether the cell has a visible nucleus or not. These performance values led us to conclude that despite the fact that STERapp cannot operate fully automatically, it can considerably reduce the analysis time; it facilitates image processing and correction and improves the precision of stereological estimations by using real areas of the objects, among other characteristics that make this software a major advance over what is currently available. In fact, the user perception of the system was evaluated using the SUS questionnaire achieving a mean score of 81.9, which means that the system is rated from good to excellent by experts. Therefore, we can conclude that STERapp is a useful software tool which can simplify and improve stereological analysis in fisheries research, specifically in those groups or institutions that use fecundity data for biological studies or fisheries assessment.

Future work will focus on the development of new algorithms to recognize the cells in the image, which improves the performance of the automatic analysis for some fish species in order to reduce the time and effort of expert supervision, perhaps taking into account these annotations into the segmentation and classification process. We would like to extend

the software to automatic recognition of objects in other stereological analysis problems coming from mineralogy or medicine.

**Supplementary Materials:** The following are available online at https://www.mdpi.com/article/10.3390/electronics10121432/s1, The user's guide of STERapp is submitted as supplementary material.

**Author Contributions:** Conceptualization, E.C., R.D.-P. and S.R.-U.; methodology, A.M., M.F.-D., E.C., M.E.G.-R. and A.F.; software, A.M., E.C., M.F.-D., M.E.G.-R. and A.F.; validation, S.R.-U., R.D.-P. and A.V.; formal analysis, A.M., E.C., M.F.-D., M.E.G.-R. and A.F.; investigation, A.M., E.C., M.F.-D., M.E.G.-R. and A.F.; resources, All; data curation, All; writing—original draft preparation, A.M., E.C. and R.D.-P.; writing—review and editing, All; visualization, All; supervision, E.C. and R.D.-P.; project administration, All; funding acquisition, All. All authors have read and agreed to the published version of the manuscript.

**Funding:** This research received no external funding.

**Institutional Review Board Statement:** Ethical review and approval was waived for this study because the fish ovary samples used in this work come from individuals caught by Spanish fishing fleet or in fisheries assessment surveys conducted under the precepts of the Common Fisheries Policy. The present work did not involve any type of experimentation with live animals.

**Data Availability Statement:** The Windows and Linux installers can be downloaded from https://citius.usc.es/transferencia/software/STERapp (accessed on 7 June 2021) for non-commercial use.

**Acknowledgments:** This work received financial support from the Xunta de Galicia (Centro singular de investigación de Galicia, accreditation 2020–2023) and the European Union (European Regional Development Fund—ERDF), Project MTM2016-76969-P, as well as from the National Program of collection, management, and use of data in the fisheries sector and support for scientific advice regarding the Common Fisheries Policy (co-funded by the European Union through the European Maritime and Fisheries Fund) in the provision of histological images of Atlantic mackerel and Roughhead grenadier.

**Conflicts of Interest:** The authors declare no conflict of interest.

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
