# Peer review of "STERapp: Semiautomatic Software for Stereological Analysis. Application in the Estimation of Fish Fecundity"

_electronics, doi:10.3390/electronics10121432_

Round 1

Reviewer 1 Report

It would be better for the paper if followings are added.

1) As a conclusion, I think quantified value of the reduction of analysis time by using STERapp have to be included. 

2) Number of data is too small for experiment and more detail explanation need to be included

3) Section 3 description is improved to include major method of research.

Author Response

See the attached document.

Reviewer 2 Report

I have recommended the manuscript 'Accept in the present form' but you can have a look at the following comments and include it in the paper.

1) Authors are suggested to include the main contributions at the end of the introduction

2) In the Results section, the performance of the proposed method should be compared with the other existing related approaches

3) Authors can include the main differences between the proposed tool and Imagej in detail.

4) The major advantages of the proposed tool over existing tools and the limitations of the proposed tool should be included.

Author Response

See the attached document.

Reviewer 3 Report

This paper describes a software for stereological analysis on images. The authors present an evaluation of the software on analysis of images of fish gonads, which indicates that using STERapp can reduce the analysis time, facilitate image processing and correction, and improve the precision of stereological estimations. 

The topic is timely and interesting. The examples discussed in the paper are practical. I am not familiar with image processing, in this review, I focus on the readability rather than the technical depth. The paper is easy to follow, and the writing is OK (still needs proofreading). My major concern is that the proposed software seems to work based on well-known existing algorithms, which lacks novelty. In addition, this work only presents a use case on fish gonads, more studies are expected to display the functions of the software in other types of image processing. 

Author Response

See the attached document.
